# Impact of Short-Term (+)-JQ1 Exposure on Mouse Aorta: Unanticipated Inhibition of Smooth Muscle Contractility

**DOI:** 10.3390/cells12111461

**Published:** 2023-05-24

**Authors:** Binjie Yan, Yu Gui, Yanan Guo, Jiaxing Sun, Mahmoud Saifeddine, Jingti Deng, Joseph A. Hill, Morley D. Hollenberg, Zhi-Sheng Jiang, Xi-Long Zheng

**Affiliations:** 1Departments of Biochemistry & Molecular Biology and Physiology & Pharmacology, Libin Cardiovascular Institute, Cumming School of Medicine, The University of Calgary, 3330 Hospital Drive N.W., Calgary, AB T2N 4N1, Canada; binjie.yan@ucalgary.ca (B.Y.);; 2Institute of Cardiovascular Disease, Key Laboratory for Arteriosclerosis of Hunan Province, Hengyang Medical College, University of South China, Hengyang 421001, China; 3Department of Physiology & Pharmacology, Cumming School of Medicine, The University of Calgary, 3330 Hospital Drive N.W., Calgary, AB T2N 4N1, Canada; 4Department of Internal Medicine (Cardiology), University of Texas Southwestern Medical Center, Dallas, TX 75390-8573, USA

**Keywords:** (+)-JQ1, smooth muscle cell, eNOS, contractility, calcium

## Abstract

(+)-JQ1, a specific chemical inhibitor of bromodomain and extraterminal (BET) family protein 4 (BRD4), has been reported to inhibit smooth muscle cell (SMC) proliferation and mouse neointima formation via BRD4 regulation and modulate endothelial nitric oxide synthase (eNOS) activity. This study aimed to investigate the effects of (+)-JQ1 on smooth muscle contractility and the underlying mechanisms. Using wire myography, we discovered that (+)-JQ1 inhibited contractile responses in mouse aortas with or without functional endothelium, reducing myosin light chain 20 (LC20) phosphorylation and relying on extracellular Ca^2+^. In mouse aortas lacking functional endothelium, BRD4 knockout did not alter the inhibition of contractile responses by (+)-JQ1. In primary cultured SMCs, (+)-JQ1 inhibited Ca^2+^ influx. In aortas with intact endothelium, (+)-JQ1 inhibition of contractile responses was reversed by NOS inhibition (L-NAME) or guanylyl cyclase inhibition (ODQ) and by blocking the phosphatidylinositol 3-kinase (PI3K)/protein kinase B (AKT) pathway. In cultured human umbilical vein endothelial cells (HUVECs), (+)-JQ1 rapidly activated AKT and eNOS, which was reversed by PI3K or ATK inhibition. Intraperitoneal injection of (+)-JQ1 reduced mouse systolic blood pressure, an effect blocked by co-treatment with L-NAME. Interestingly, (+)-JQ1 inhibition of aortic contractility and its activation of eNOS and AKT were mimicked by the (−)-JQ1 enantiomer, which is structurally incapable of inhibiting BET bromodomains. In summary, our data suggest that (+)-JQ1 directly inhibits smooth muscle contractility and indirectly activates the PI3K/AKT/eNOS cascade in endothelial cells; however, these effects appear unrelated to BET inhibition. We conclude that (+)-JQ1 exhibits an off-target effect on vascular contractility.

## 1. Introduction

The family of bromodomain (BRD) and extra-terminal domain (BET) proteins comprises four members—BRD2, BRD3, BRD4, and bromodomain testis-specific protein (BRDT)—and plays a critical role in the development and pathogenesis of cardiovascular diseases (CVDs), including heart failure, atherosclerosis, pulmonary arterial hypertension, and renal ischemia-reperfusion injury [1,2,3,4,5]. (+)-JQ1 is a small molecule that selectively inhibits the BET bromodomain by competing for the acetyl-lysine binding pocket and displacing the BET protein from chromatin, which alters the transcriptional activity of the target gene [6]. While (+)-JQ1 was initially developed as a BET protein inhibitor, most studies have shown that its effect is mainly due to the selective inhibition of BRD4. For example, (+)-JQ1 was reported to increase p21 expression, thus triggering cell cycle arrest of pulmonary artery (PA) smooth muscle cells from sugen/hypoxia pulmonary arterial hypertensive rats [2]. Molecular (siRNA) inhibition of BRD4 had the same effect as (+)-JQ1 on SMC phenotype and restoring mitochondrial membrane potential [2]. Wang et al. reported that, in a rat balloon angioplasty model, local application of (+)-JQ1 but not the inactive enantiomer (−)-JQ1 diminished neointima of carotid arteries through inhibition of BRD4 [3]. These effects involve downregulation of platelet-derived growth factor (PDGF) receptor-α [3].

More recently, Dutzmann et al. reported that local application of (+)-JQ1 significantly attenuated neointimal lesion of the femoral artery in C57BL/6 mice in response to wire-induced injury [7]. These authors proposed this effect was through arresting cells in the G0/1 phase, inhibiting SMC proliferation without inducing apoptosis by targeting BRD4, and also revealed (+)-JQ1 interruption of direct binding of BRD4 to FOXO1, which subsequently reduced FOXO1 transactivational capacity [7]. In summary, (+)-JQ1 has been reported to inhibit vascular SMC proliferation and its local application ameliorates neointima formation in rat carotid and mouse femoral arteries in response to wire injury. However, how short-term (+)-JQ1 exposure regulates smooth muscle contractility has not been tested. It was reported that (+)-JQ1 treatment suppresses phosphorylation and activation of eNOS in vascular endothelial growth factor (VEGF)-stimulated HUVECs [8]. In spontaneously hypertensive rats (SHR), (+)-JQ1 treatment increases eNOS activity [9]. Thus, it is possible that (+)-JQ1 regulates the contractility of vascular smooth muscle through its modulation of eNOS activity.

In the present study, we demonstrate that (+)-JQ1 administered through intraperitoneal injection ameliorates mouse carotid arterial neointima formation induced by complete carotid ligation. Importantly, (+)-JQ1 inhibits contractile responses of mouse aortas through activating eNOS in endothelial cells. Interestingly, (+)-JQ1 also directly inhibits smooth muscle contraction without functional endothelium. It is critical to note that (−)-JQ1 has the same effects on smooth muscle contractility, and knockout (KO) of BRD4 in mouse SMCs does not affect (+)-JQ1’s inhibitory effect. In conclusion, (+)-JQ1 exhibits BET protein-independent inhibition of smooth muscle contractility through both endothelium-dependent and -independent mechanisms.

## 2. Methods and Materials

### 2.1. Materials

Human umbilical vein endothelial cells (C2519AS), pooled donor, were purchased from the Lonza Walkersville (Walkersville, MD, USA). Cre recombinase adenovirus (Ad-Cre-GFP; No. 1700) and Null adenovirus (Ad-Null-GFP; No. 1240) were purchased from Vector Biolabs (Malvern, PA, USA). Endothelial cell medium (#1001) was purchased from ScienCell (Carlsbad, CA, USA) 10,000 units/mL penicillin (#15140122), and 10,000 units/mL streptomycin solution (#15140122) were purchased from Thermo Fisher Scientific (Ottawa, ON, Canada). (+)-JQ1 and (−)-JQ1 were purchased from Adooq (Irvine, CA, USA). *N*^ω^-nitro-L-arginine methyl ester hydrochloride (L-NAME), 1H- [1,2,4] oxadiazole [4,3-a] quinoxaline-1-one (ODQ) and ketoconazole (KNZ) were purchased from Cayman Chemical (Ann Arbor, MI, USA). Actinomycin D (ActD), AKT inhibitor IV (AKTi IV), bafilomycin A1 (Baf.A1), cycloheximide (CHX), chloroquine (CQ), calcium ionophore (A23187), indomethacin, LY294002, potassium chloride (KCl), (R)-(-)-phenylephrine hydrochloride (PE), and acetylcholine chloride (Ach) were purchased from Sigma-Aldrich (St. Louis, MO, USA). QNZ (EVP4593) was purchased from Selleckchem (Houston, TX, USA). TFLLR-NH2, prazosin and SKF-96365 were purchased from MedChem Express (Monmouth Junction, NJ, USA). Antibodies including phospho-myosin light chain 20 (Ser19) #3671, myosin light chain 20 #8505, phospho-eNOS (Ser1177) #9571, phospho-AKT (Ser 473) #4060, pan-AKT (C67E7) #4691, anti-mouse IgG HRP-linked secondary antibody #7076 and anti-rabbit IgG HRP-linked secondary antibody #7074 were purchased from Cell Signaling Technology (Danvers, MA, USA). Total-eNOS (ab76198) was purchased from Abcam (Cambridge, MA, USA). Calponin **#**C2687 and β-actin #A5441 were purchased from Sigma-Aldrich (St. Louis, MO, USA).

### 2.2. Cell Lines and Cell Culture

HUVECs were cultured in an endothelial cell medium supplemented with fetal bovine serum (FBS) and endothelial cell growth supplement, and incubated at 37 °C in an atmosphere of 5% CO_2_. The medium also included 100 U/mL penicillin and 100 μg/mL of streptomycin and was changed every 48 h.

### 2.3. Primary Culture of Mouse Aortic SMCs

Primary culture of SMCs was performed from the aortas of 8-week-old male C57BL/6J mice using an enzyme dispersion method, as we previously described [10]. In brief, the aortas were minced and treated with collagenase (1.5 mg/mL) for 3 h, after which the cells were isolated and cultured in a nutrient medium containing 10% FBS, 100 U/mL penicillin, 100 μg/mL streptomycin, and 2 mmol/L L-glutamine. The cells were grown in a 37 °C incubator with 5% CO_2_ and used for experimentation when they were in passages 4 to 10.

### 2.4. Western Blot Analysis

Cultured HUVECs or mouse aortic tissues were washed with cold PBS three times followed by lysis with RIPA buffer (150 mM sodium chloride, 1.0% NP-40; 0.5% sodium deoxycholate, 0.1% sodium dodecyl sulfate, 50 mM Tris pH 8.0, supplemented with protease inhibitors) for 30 min on ice. Aortic tissues were lysed in RIPA buffer for 2 h under agitation, following homogenization with mortar and pestle. Samples were centrifuged at 14,000× *g* for 15 min at 4 °C, and the supernatant was kept for analyses. Total proteins were quantified using the measurement of their concentrations with an iMark™ Microplate Reader (Bio-Rad Laboratories, Hercules, CA, USA). The same amount of protein per sample was separated by 10–12% sodium dodecyl sulfate-polyacrylamide gel electrophoresis (SDS-PAGE) and then transferred to nitrocellulose filter membranes. Before immunodetection, membranes were blocked within 10% non-fat dry milk and then incubated with primary antibodies (1:1000) overnight at 4 °C. Subsequently, membranes were incubated with a secondary antibody (1:10,000) for 1 h at room temperature after washing with TBST (Tris-buffered saline with 0.1% Tween^®^ 20 detergent). The abundance of immunolabelled proteins was detected using the enhanced chemiluminescence (ECL) detection system, acquired by ImageQuant LAS-4000 (GE Healthcare, Mississauga, ON, Canada), and quantitated using Gel-Pro Analyzer version 4.0 software (Media Cybernetics, Maryland, MD, USA). Each experiment was repeated in triplicate.

### 2.5. Myography

C57BL/6J male mice were purchased from the Jackson Laboratory (6- to 8-week-old, 19–20 g). All animal studies were approved by the Institutional Animal Care and Use Committees at the University of Calgary and were performed in accordance with the US National Institutes of Health guidelines. Mice were euthanized by decapitation under halothane anesthesia, followed by removal of the thoracic aortas, which were immersed at 37 °C in Krebs solution containing the following: 114 mM NaCl, 4.7 mM KCl, 0.8 mM KH_2_PO_4_, 1.2 mM MgCl_2_, 11 mM D-glucose, 25 mM NaHCO_3_, and 2.5 mM CaCl_2_, pH 7.4. The thoracic aortas were dissected and cut into rings (2.5 mm in length), mounted in 5-mL myograph chambers (Danish Myo Technology, Skejbyparken, Denmark) by two metal hooks, and maintained in Krebs solution bubbled with a 95% O_2_/5% CO_2_ gas mixture at a resting tension of 0.5 g (4.9 mN). Isometric tension was recorded using MyoDaq/MyoData 2.1 software (Danish Myo Technology, Skejbyparken, Denmark). After equilibration, each ring was contracted with 1 µM PE, followed by relaxation induced by 1 µM ACh. Vessel segments displaying >50% relaxation to ACh were considered to have intact endothelium. The aortic ring was perfused with a 1 mg/mL solution of sodium deoxycholate in saline for 30 s to remove the vascular endothelium, as described previously [11]. 

### 2.6. Ca^2+^ Flux in Isolated Aortic Rings

To investigate the inhibitory effect of (+)-JQ1 on extracellular Ca^2+^ influx, extracellular and intracellular calcium were eluted by Ca^2+^-free Krebs solution with 1 mmol/L EGTA. The aortic rings were incubated with 10 μmol/L (+)-JQ1 for 20 min, and then 50 mM KCl or 1 μmol/L PE was added to the organ bath and co-incubated for 10 min. Subsequently, a final concentration of 2.5 mmol/L CaCl_2_ solution was added to the water bath. The KCl or PE in the bath caused a Ca^2+^ influx into the SMCs and vasoconstriction. The vascular tone was recorded to reflect the extracellular calcium influx levels.

### 2.7. Ca^2+^ Signal Assay

Calcium signaling experiments were performed as previously described [12]. Briefly, mouse primary aortic SMCs were grown to approximately 80% confluency in a T75 flask and detached using an enzyme-free cell dissociation buffer. The cells were then washed with PBS and incubated with the calcium indicator dye Fluo-4-AM NW for 45 min at room temperature. The increase in fluorescence emission at 530 nm was used as an indicator of intracellular calcium levels. Calcium signaling was monitored using an Aminco Bowan series II fluorimeter and the AB2 V5.10 software. When appropriate, the responses were normalized to the calcium signal generated by the calcium ionophore A23178 (2.5 µM).

### 2.8. Generation of Mice for Conditional KO of BRD4 in SMCs

The BRD4 flox line was generated by Dr. Hill’s lab at University of Texas Southwestern [4], and was bred with male mice harboring a transgene of *SMMHC-CreERT2* in the Y chromosome (*Myh11Cre/+*, The Jackson Laboratory: 019079, C57BL/6J strain) to generate male *Myh11Cre/+*: *BRD4^flox/flox^* mice. Given that the *Cre* gene is inserted in the Y chromosome, only male mice express the Cre protein, and Cre-adenovirus induces KO of the BRD4 gene only in male mice. Mouse aortic rings with denuded endothelium from male *Myh11Cre/+*: *BRD4^flox/flox^* mice were cultured with Ad-Null-GFP or Ad-Cre-GFP for 3 days to specifically knock out *BRD4* in SMCs. Then, these tissues were used for myography. Before myography, vascular perfusion with a solution of sodium deoxycholate was used to remove the vascular endothelium.

### 2.9. Non-Invasive Blood Pressure Measurement following (+)-JQ1 Administration 

Before the non-invasive blood pressure test, 12 male mice were randomly divided into 3 groups: DMSO, (+)-JQ1 (10 mg/kg/day), (+)-JQ1 (10 mg/kg/day) plus L-NAME (25 mg/kg/day) [13], via intraperitoneal injection for 14 days. PE (100 µg/kg) [13] was injected intraperitoneally 30 min in advance of experiments. C57BL/6J male mice were placed in the prewarmed tail-cuff device once a day for 5 consecutive days to acclimatize them to the procedure. We used a computerized tail-cuff system to measure heart rate and systolic/diastolic/mean blood pressure [14]. 

### 2.10. Neointima Formation in Mouse Carotid Arteries

Male C57BL/6J mice, approximately 18 weeks old, were randomly assigned to either a vehicle control group or a (+)-JQ1 treatment group (with the investigator blinded to the allocation). The mouse left carotid artery (LCA) was completely dissected and ligated proximal to the bifurcation of the internal and external carotid arteries, as previously described [10]. Male mice received daily treatment with 10 mg/kg JQ1 (i.p.), while the control group received vehicle treatment for 3 weeks. On day 21 post-ligation, the animals were euthanized using a CO_2_ chamber and subsequently perfused. Carotid arteries were then isolated, and arterial tissues were fixed in 4% paraformaldehyde at 4 °C overnight before being embedded in paraffin. Serial 5 μm-thick paraffin sections were prepared, covering a region 2 mm proximal to the ligation site, and were subsequently stained with hematoxylin and eosin (HE).

### 2.11. Statistical Analysis

Data are representative of at least three independent experiments unless otherwise stated. Data are presented as mean ± standard deviation (SD). Statistical analysis was performed with IBM SPSS Statistics version 25.0 (IBM, Chicago, IL, USA) using one-way analysis of variance (ANOVA), or an unpaired two-tailed *t*-test (Student’s *t*-test). For each statistical test, a *p*-value of < 0.05 was considered significant.

## 3. Results

### 3.1. (+)-JQ1 Attenuates Mouse Aortic Contractile Responses by KCl and PE

Previous studies have shown that (+)-JQ1 inhibits SMC proliferation both in vivo and in vitro [2,3]. In this study, we investigated the effects of (+)-JQ1 on smooth muscle contractility using tissue wire myography. To achieve this, we tested KCl and PE in our bioassay, both of which induce smooth muscle contraction through voltage-dependent calcium channels and G-protein coupled receptors, respectively. Initially, we examined the impact of varying concentrations of (+)-JQ1 on KCl- and PE-induced contractile responses in mouse aortic rings, with and without functional endothelium. Aortic preparations with intact endothelium, as indicated by the presence of >50% Ach-induced relaxation, were pretreated with different concentrations of (+)-JQ1 (0.1, 1, 10, 50, and 100 μM) for 20 min, followed by the addition of KCl (50 mM) or PE (1 μM). The absence of endothelium was indicated by a lack of Ach-induced relaxation. The representative tracings are shown in Figure 1A–D. Our cumulative data have shown that the inhibitory effect of (+)-JQ1 was concentration-dependent, with significant inhibition observed at 10 µM, which reduced about 42% of contractile responses by KCl (Figure 1E) and up to 60% by PE (Figure 1F) with functional endothelium. Without functional endothelium, (+)-JQ1 reduced approximately 40% of contractile responses by KCl (Figure 1G) and 46% by PE (Figure 1H). For KCl-induced contraction, (+)-JQ1 at 50 or 100 μM induced more significant inhibition in endothelium-intact preparations compared with those without functional endothelium (Figure 1I). For PE-induced contraction, endothelium-intact preparations were more sensitive to (+)-JQ1 inhibition, as indicated by significantly more inhibition by (+)-JQ1 at 10 μM (Figure 1J). Note that this inhibitory effect was reversible within 30–60 min after washout of (+)-JQ1 (Appendix A), and addition of (+)-JQ1 at the peak contraction by KCl or PE also reduced the tension as a vasodilator (Appendix A).

### 3.2. (+)-JQ1 Inhibition of Aortic Contraction in Endothelium Intact Preparations Involves Activation of eNOS and Guanylyl Cyclase (GC)

We then determined the mechanism underlying the acute inhibitory effects of (+)-JQ1 on contractile responses induced by KCl and PE. We first examined the involvement of eNOS activation and NO release, and NO-sensitive GC on smooth muscle. Our results showed that co-treatment with L-NAME, a well-known inhibitor of NOS [15], or ODQ, an inhibitor of NO-stimulated GC [16], significantly reversed (+)-JQ1 inhibition of contractile responses induced by KCl (Figure 2A) and PE (Figure 2B). The representative tracings for PE are shown in Figure 2C, in which the presence of endothelium was indicated by Ach-induced relaxation. Note that co-treatment with L-NAME or ODQ completely reversed the PE-induced, but not KCl-induced, contraction in endothelium-intact mouse aortic preparations (Figure 2A,B). In aortic preparations without functional endothelium, however, neither L-NAME nor ODQ significantly affected contractile responses of KCl and PE (Figure 2D–F). 

Pretreatment with indomethacin, an inhibitor for cyclooxygenase, did not affect (+)-JQ1-induced inhibition in any preparation (Appendix A). In addition, pretreatment with ActD (1 μM) or CHX (10 μM) did not affect (+)-JQ1-induced contraction by KCl or PE (Appendix A), suggesting that inhibition of contractile responses by (+)-JQ1 in either endothelium-intact mouse aortic rings or the preparations without functional endothelium did not involve any gene transcription or protein translation.

In sum, our data from pharmacological studies suggest that (+)-JQ1 induces acute inhibition of contractile response of mouse aortas through activating endothelial eNOS with a functional endothelium, and a direct effect on smooth muscle when endothelium is denuded.

### 3.3. (+)-JQ1 Attenuation of Contractile Responses Is Calcium-Dependent, and through Inhibiting the Contractile Machinery

To further confirm the inhibition of smooth muscle contraction of mouse aortas with and without functional endothelium, we determined the dependence of (+)-JQ1 inhibitory effects on extracellular Ca^2+^. Our results showed that the inhibition of contractile responses in (+)-JQ1 in both preparations with (Figure 3A) and without (Figure 3B) functional endothelium was dependent on the concentration of extracellular Ca^2+^. More specifically, (+)-JQ1 significantly inhibited PE-induced contraction in the presence of 1 μM Ca^2+^, but not 0.1 μM, in the preparations with functional endothelium (Figure 3A). As a comparison, in the absence of functional endothelium, (+)-JQ1 significantly inhibited PE-induced contraction even in the presence of 0.1 μM Ca^2+^ (Figure 3B).

Thus, we examined the phosphorylation of LC20 in response to PE treatment in both preparations in the presence or absence of (+)-JQ1. As shown in Figure 3C,D, phosphorylation levels of LC20 were significantly downregulated by (+)-JQ1 treatment, which was completely reversed by co-treatment with either L-NAME or ODQ. In mouse aortic rings without functional endothelium, (+)-JQ1 also significantly reduced phospho-LC20 levels, which were not significantly affected by co-treatment with either ODQ or L-NAME (Figure 3E,F). All these data indicated that (+)-JQ1 inhibition of smooth muscle contraction shares the final, common mechanism that converges at phosphorylation of LC20 in SMCs, and in mouse aorta with or without intact endothelium.

### 3.4. (+)-JQ1 Inhibits Ca^2+^ Influx in Cultured SMCs

Our myography data suggested a direct inhibitory effect of (+)-JQ1 on Ca^2+^ influx in SMCs. To verify this possibility, we performed Ca^2+^ signal assays in primary cultured mouse aortic SMCs. Interestingly, a high concentration of PE (50 μM) induced an increase in intracellular Ca^2+^, which was reduced by pre-incubation with 50 μM (+)-JQ1 but not by α receptor antagonist, prazosin at 5 μM (Appendix A). Thus, we tested the effects of (+)-JQ1 on Ca^2+^ responses induced by other agonists, such as TFLLR-NH2 (TF), the proteinase-activated receptor 1 (PAR1) agonist, which is known to induce intracellular Ca^2+^ release [17,18]. As shown in Figure 4A,B, pretreatment with (+)-JQ1 (50 µM) did not affect the TF-induced increase in intracellular Ca^2+^ in primary cultured mouse aortic SMCs. Thereafter, we examined the effects of (+)-JQ1 on Ca^2+^ induced by calcium ionophore A23187 (CI), which is known to induce Ca^2+^ influx across cell membranes [19]. Note that, in the Ca^2+^-free buffer (Appendix A), the addition of CI induced a rapid, transient increase in intracellular Ca^2+^, which resulted from the mobilization of intracellular stores [19]. In the buffer containing Ca^2+^ (Appendix A), however, CI treatment induced a bi-phase Ca^2+^ effect, namely, a rapid transient increase in Ca^2+^ followed by a sustained increase in Ca^2+^ signal, resulting from an influx of extracellular Ca^2+^ [19]. Importantly, pretreatment with (+)-JQ1 (50 µM) significantly reduced the sustained increase in Ca^2+^, but not the rapid, transient increase in Ca^2+^, induced by CI (Figure 4C,D) and CI (Appendix A). Notably, the effect of (+)-JQ1 on the Ca^2+^ signal was mimicked by SKF 96365 (20 µM, 5 min), a well-established Ca^2+^ channel blocker [19] (Figure 4E,F). Together, our data indicate that (+)-JQ1 has a direct inhibitory effect on Ca^2+^ influx in aortic SMCs.

### 3.5. Direct Inhibition of Smooth Muscle Contraction by (+)-JQ1 Is Independent of BRD4 in SMCs

As discussed in the introduction, (+)-JQ1 has been widely used to characterize the role of BRD4 in various biological functions. The rapid effects (within several minutes) we observed do not support the hypothesis that (+)-JQ1 inhibits BRD4 and exerts its effects through an epigenetic mechanism. To examine the role of BRD4 in (+)-JQ1-induced inhibition of mouse aortic contraction, we measured the effects of (+)-JQ1 on the contractility of mouse aortas with BRD4 knockout. As described in Methods, the aortas from *BRD4* gene floxed mice were isolated for organ culture with the presence of Ad-Cre-GFP or Ad-Null-GFP, followed by WB confirmation of BRD4 knockout (Figure 5A). Our further investigations revealed that (+)-JQ1 continued to inhibit smooth muscle contraction in aortic tissues, even in the absence of intact endothelium and in the presence of BRD4 knockout (Figure 5B–D). Our data suggest that the inhibition of smooth muscle contraction by (+)-JQ1 is independent of smooth muscle BRD4. 

### 3.6. (+)-JQ1 Activates eNOS through the PI3K/AKT Pathway in Endothelial Cells

In subsequent studies, we focused on how (+)-JQ1 activates eNOS in endothelium cells (ECs). It is well known that activation of the PI3K/AKT pathway results in eNOS activation in ECs. To determine this possibility, we examined whether inhibition of PI3K or AKT reversed (+)-JQ1-induced inhibition of contractility. As shown in Figure 6A, co-treatment with PI3K inhibitor, LY-29004 (5 μM), or AKT inhibitor, AKTi IV (5 μM), partially but significantly reversed the inhibitory effect of (+)-JQ1 on PE-induced contraction. All these data suggested that (+)-JQ1 activates eNOS through the PI3K/AKT pathway. 

To further establish this mechanism, we took advantage of cultured HUVECs and examined the phosphorylation levels of AKT and eNOS in response to (+)-JQ1 treatment. Our results showed that (+)-JQ1 at 10 μM time-dependently increased the phosphorylation of ATK at the Ser473 site (Figure 6B,C) and eNOS at the Ser1177 site (Figure 6B,D). Notably, a significant increase in phospho-AKT and phospho-eNOS was observed as early as 5 min. Importantly, co-treatment with LY-29004 (1 μM) or AKTi IV (1 μM) abolished the phosphorylation of AKT in response to treatment with (+)-JQ1 (10 μM, Figure 6E,F) and also significantly reduced eNOS phosphorylation induced by (+)-JQ1 (Figure 6E,G). There was no significant change in BRD4 protein abundance during the short treatment with (+)-JQ1 as detected by WB (Figure 6E).

Given that the evidence suggests a role for BRD4 in the activation of NF-κB and autophagy [20], we examined the potential involvement of NF-κB and autophagy in (+)-JQ1-induced inhibition of smooth muscle contractility. Our results showed that co-treatment with QNZ (NF-κB inhibitor, 10 µM), rapamycin (autophagy activator, 100 nM), Baf.A1 (100 nM), and CQ (autophagy inhibitors, 30 μM) did not significantly affect (+)-JQ1-induced inhibition of smooth muscle contractility (Appendix A).

### 3.7. Inhibition of (+)-JQ1 Metabolism by Ketoconazole (KNZ) Does Not Affect (+)-JQ1-Induced Inhibitory Effects on KCl and PE-Induced Contractile Responses

Due to the rapid metabolism of (+)-JQ1, its half-life is only about 1 h in CD1 mice that are genetically engineered to express CD1 proteins [6]. Recently, it was reported that (+)-JQ1 could be metabolized in the liver by a cytochrome P450 family 3 subfamily A member 4 (CYP 3A4) [21], which is expressed in vascular endothelial cells [22], suggesting that the metabolites from (+)-JQ1 may mediate its effects on vascular contractility, such as activation of PI3K/AKT and eNOS. To test this possibility, we pretreated aortic tissues with intact endothelium with KNZ, a CYP3A4 inhibitor, since CYP3A4 is the main contributor to the production of (+)-JQ1 metabolites in vitro [21]. Our results showed that KNZ (1 µM), which alone did not affect aortic contractility, did not significantly reverse the (+)-JQ1-induced inhibitory effects on contractile responses induced by KCl or PE (Appendix A). These results suggest that the acute inhibitory effect of (+)-JQ1 in isolated mouse aorta is independent of metabolism.

### 3.8. The Inhibitory Effect of (+)-JQ1 Is Mimicked by (−)-JQ1

(+)-JQ1 binds to the bromodomain of BET proteins and prevents them from binding to specific regions of chromatin, thereby disrupting the normal regulation of gene expression. To date, many studies demonstrated that (−)-JQ1, an enantiomer of (+)-JQ1, does not have the same pharmacological effects on BET proteins as (+)-JQ1 [6,23]. Thus, we further examined the effect of (−)-JQ1 on vascular contractility. As shown in Figure 7, (−)-JQ1 had the same effect on PE-induced contraction in mouse aortas as (+)-JQ1 did (Figure 7). Similarly, co-treatment with L-NAME (100 µM) or ODQ (10 μM) also reversed the (−)-JQ1-induced inhibitory effects on PE-induced contractile responses (Figure 7A). Like (+)-JQ1, addition of (−)-JQ1 at the peak of contraction by KCl or PE also reduced the tension as a vasodilator (Appendix A). In cultured HUVECs, (−)-JQ1 mimicked the stimulatory effect of (+)-JQ1 on the activation and phosphorylation of AKT (Figure 7B,C) and eNOS (Figure 7B,D).

### 3.9. (+)-JQ1 Reduces Systolic Blood Pressure in Mice

To further verify the inhibitory effect of (+)-JQ1 on vascular smooth muscle contractility in vivo, we measured the blood pressure of mice treated with DMSO, (+)-JQ1 or (+)-JQ1 plus L-NAME for up to 14 days using the tail-cuff method as described in Methods. Our results showed that (+)-JQ1 significantly reduced systolic blood pressure (Figure 8A) but not diastolic and mean blood pressure (Figure 8B,C). Co-treatment with L-NAME blocked the (+)-JQ1-induced decrease in systolic blood pressure (Figure 8A), suggesting a role of nitric oxide released in response to (+)-JQ1.

### 3.10. (+)-JQ1 Administered via Intraperitoneal Injection Ameliorates Neointima Formation in Mouse Carotid Arteries Induced by Complete Carotid Ligation

Previous studies demonstrated that local application of (+)-JQ1 reduces neointima formation in rat carotid and mouse femoral arteries induced by mechanical injury [2,3]. Our findings prompted us to investigate whether (+)-JQ1, when given intraperitoneally, inhibits neointima lesions in mouse carotid arteries induced by complete carotid ligation. Indeed, our results revealed that (+)-JQ1 significantly ameliorates neointima lesions (Figure 9). In our in vitro studies, (+)-JQ1 not only inhibited BrdU incorporation and proliferation of rat aortic SMCs but also induced apoptosis, as shown in Appendix A.

## 4. Discussion

Our study revealed that (+)-JQ1 can inhibit the contractile responses of mouse aortas, reduce LC20, and block Ca^2+^ influx in primary cultured SMCs. In aortas with an intact endothelium, (+)-JQ1’s inhibition of contractile responses was reversed by NOS or GC inhibition, and by PI3K/AKT pathway inhibition. Activation of eNOS by (+)-JQ1 was further validated in cultured HUVECs). Additionally, intraperitoneal injection of (+)-JQ1 also reduced mouse systolic blood pressure and neointima lesions in mouse carotid arteries induced by complete carotid ligation. However, (+)-JQ1’s effects on smooth muscle contractility were mimicked by (−)-JQ1 and were not modulated by BRD4 deficiency in SMCs, suggesting that (+)-JQ1 has an off-target, BET protein-independent effect. These findings are novel and significant given that (+)-JQ1 has been used in clinical trials and has been implicated in the inhibition of SMC proliferation and neointima lesion formation through BRD4, as indicated in previous studies.

First, we have demonstrated that (+)-JQ1 inhibits the contractile responses of mouse aortas with functional endothelium through the downregulation of LC20 phosphorylation and dependence on extracellular Ca^2+^. Inhibiting PI3K, AKT, eNOS, or GC activation reversed (+)-JQ1’s inhibition of aortic contraction, suggesting the involvement of eNOS activation. This possibility was confirmed with cultured HUVECs, in which treatment with (+)-JQ1 rapidly increased phosphorylation of AKT and eNOS. LY-29004 and AKTi IV, which inhibit PI3K and AKT, respectively, blocked this effect. These findings support our conclusion that (+)-JQ1 stimulates eNOS activity, which was further confirmed by our in vivo study, in which intraperitoneal injection of (+)-JQ1 reduced mouse systolic blood pressure that was blocked by co-treatment with L-NAME. Our findings are consistent with previous observations that (+)-JQ1 stimulates eNOS in SHR, but opposite to the finding that (+)-JQ1 inhibits eNOS in HUVECs [8,9]. The activation of eNOS by (+)-JQ1 may have therapeutic implications since NO, primarily produced by eNOS in the vascular endothelium, plays a crucial role in regulating blood pressure. Endothelial dysfunction, characterized by reduced bioavailability of NO, is a key factor contributing to the elevation of blood pressure [24,25]. 

Another interesting discovery in our study was that (+)-JQ1 directly inhibits contraction in mouse aortas without endothelium, indicating a direct inhibitory effect on smooth muscle contraction. Importantly, this inhibition by (+)-JQ1 was also through the downregulation of LC20 phosphorylation and dependence on extracellular Ca^2+^, but was not affected by any inhibitors for PI3K, AKT, eNOS, or GC. In our calcium signaling assay with primary cultured mouse aortic SMCs, our results showed that pretreatment with (+)-JQ1 significantly suppressed the influx of extracellular Ca^2+^, which provides some insight into the mechanism underlying the direct inhibition of (+)-JQ1 on smooth muscle contraction. However, the exact mechanism remains unclear and requires further investigation.

Importantly, our data suggest that the effects of (+)-JQ1 on endothelial cells and smooth muscle are off-target effects. First, (−)-JQ1, a compound structurally incapable of BET inhibition, mimics the effects of (+)-JQ1 on contractile responses and activation of AKT and eNOS in HUVECs, indicating that the effects of (+)-JQ1 were independent of BET inhibition. Our data from mouse aortas with BRD4 KO support the conclusion that (+)-JQ1-induced direct inhibition of mouse aortic smooth muscle contractility is independent of smooth muscle BRD4. Indeed, based on the required time of 10–20 min for (+)-JQ1’s inhibitory effect, it is conceivable that its effects are independent of gene transcription or protein translation, which was confirmed by our studies using ActD and CHX, inhibitors for gene transcription and protein translation, respectively. In addition, the inhibitory effect of (+)-JQ1 on mouse aortic contractility may be independent of its metabolism. It is known that (+)-JQ1 metabolism occurs in human and mouse liver microsomes, and CYP3A4 has been identified as the main contributor to the production of (+)-JQ1 metabolites in vitro [21]. Therefore, the metabolites of (+)-JQ1 may mediate (+)-JQ1’s effects in vivo and in vitro. However, our study showed that the CYP3A4/5 inhibitor KNZ, which strongly inhibits (+)-JQ1 metabolism in both human and mouse liver microsomes [21], did not significantly affect (+)-JQ1’s inhibition of smooth muscle contractility.

Nevertheless, our conclusion has been supported by several published studies. In prostate cancer, for example, (+)-JQ1, but not (−)-JQ1, significantly inhibited cancer growth but promoted invasion and metastasis through its direct interaction and inhibition of forkhead box protein A1 (FOXA1), an invasion suppressor in prostate cancer, independent of any BET proteins [23]. To our surprise, this study showed that (−)-JQ1 does not mimic (+)-JQ1 effects. In contrast, our study has, for the first time, shown that (+)-JQ1 has off-target, BET protein-independent effects on the contractility of vascular smooth muscle, which are mimicked by (−)-JQ1. It is also worth mentioning that our data demonstrate that (+)-JQ1, when administered via intraperitoneal injection, significantly ameliorates neointima lesions caused by complete carotid ligation, which is consistent with previous findings by others [2,3]. However, the results from the current study suggest that (+)-JQ1’s effects on SMCs, both in vivo and in vitro, are off-target effects and independent of BRD4 or other BET proteins. 

In summary, our results demonstrate that (+)-JQ1 inhibits smooth muscle contractility of mouse aortas through both endothelium-dependent and -independent mechanisms. In aortic preparations with intact endothelium, (+)-JQ1 stimulates the PI3K/AKT pathway and subsequently activates eNOS to exert its inhibitory effect on smooth muscle contractility. (−)-JQ1 mimics (+)-JQ1’s effects on eNOS activation in vitro and directly inhibits smooth muscle contractility. Therefore, we conclude that (+)-JQ1 has off-target effects on vascular smooth muscle contractility, which may have therapeutic implications. Since (+)-JQ1, as an inhibitor of the BET protein family, has been widely used to treat various diseases [3,26,27,28,29,30], it is important to consider our findings for data interpretation in all future studies involving (+)-JQ1.

## 5. Conclusions

(+)-JQ1 inhibits smooth muscle contractility in mouse aortas through a dual mechanism involving activation of eNOS and inhibition of Ca^2+^ influx in SMCs. (−)-JQ1 mimics (+)-JQ1’s effects on eNOS activation in vitro and directly inhibits smooth muscle contractility (Figure 10). BRD4 deficiency did not modulate (+)-JQ1’s inhibition of smooth muscle contractility. It is crucial to consider these findings when interpreting data in all future studies involving (+)-JQ1.

## Figures and Tables

**Figure 1 cells-12-01461-f001:**
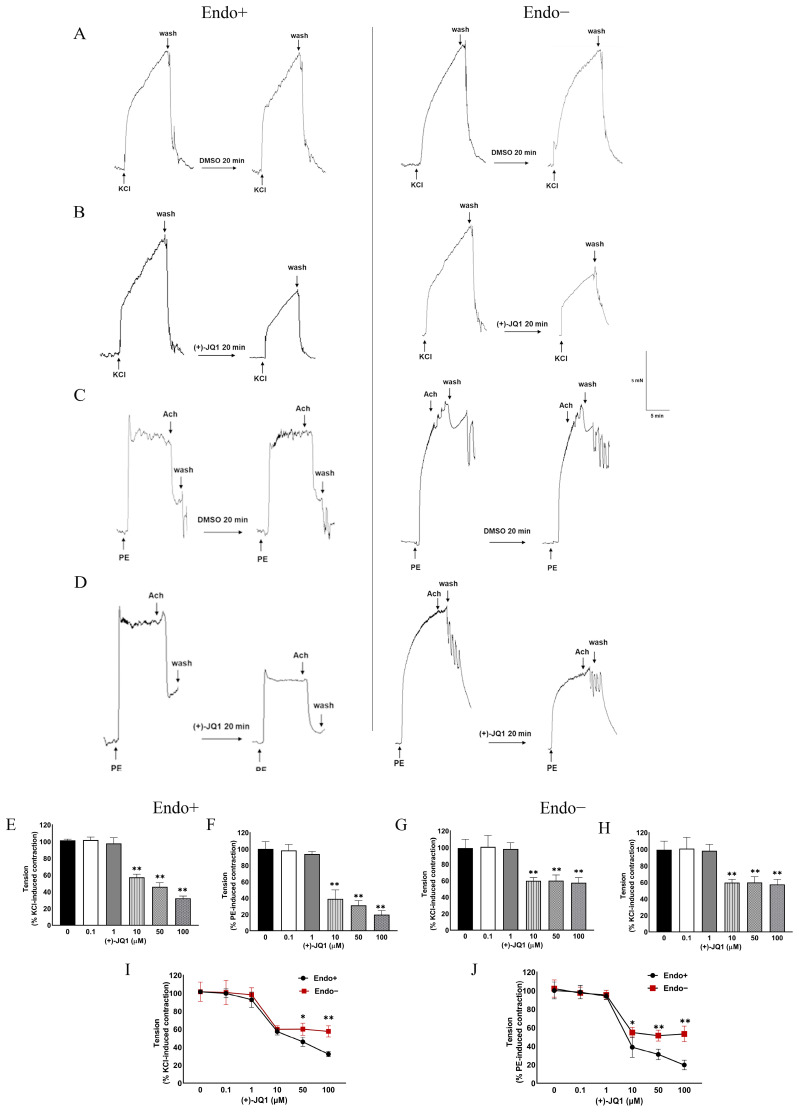
(+)-JQ1 inhibits KCl- and PE-induced contraction in mouse aortic tissues with or without endothelium. (**A**–**D**): Representative recordings. Mouse aortas with (left panel) or without (right panel) an intact endothelium were pretreated with DMSO or (+)-JQ1 (10 µM) for 20 min, followed by exposure to KCl (50 mM, **A**,**B**) or PE (1 µM, **C**,**D**). (**E**–**H**): Cumulative data showing that the inhibitory effect of (+)-JQ1 (0, 0.1, 1, 10, 50 or 100 µM) on the contraction induced by KCl or PE in endothelium-intact (Endo+) or denuded (Endo−) mouse aortic artery rings. (**I**,**J**): (+)-JQ1 shows different inhibitory effects between Endo+ and Endo− mice aortic rings pre-contracted by KCl (50 mM) and PE (1 µM). * *p* < 0.05; ** *p* < 0.01; *n* = 6.

**Figure 2 cells-12-01461-f002:**
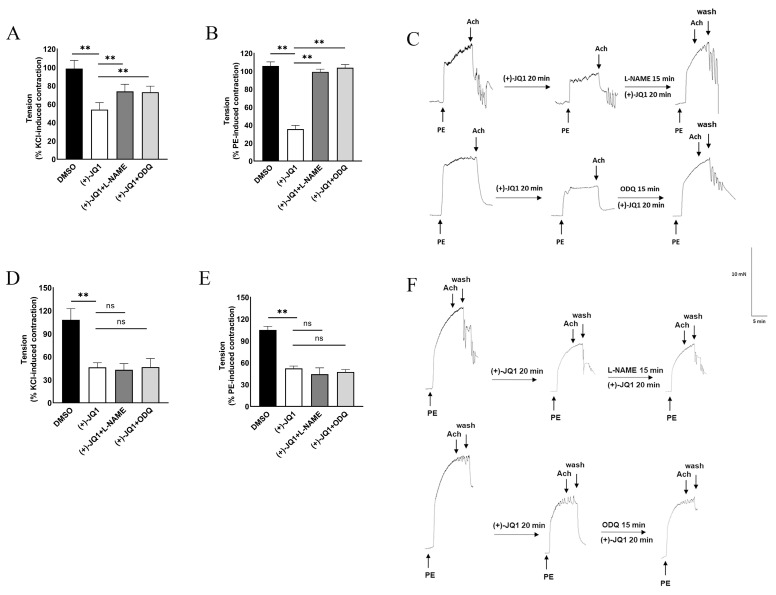
Pretreatment with L-NAME or ODQ reverses inhibitory effects of (+)-JQ1 on KCl- and PE-induced contraction in the presence or absence of endothelium. (**A**,**B**): Cumulative data showing the effects of L-NAME and ODQ on the (+)-JQ1-induced inhibition of contractility with functional endothelium. ** *p* < 0.01; *n* = 4. (**C**): Representative recordings: Mouse aortas with intact endothelium were pretreated with L-NAME (100 µM) or ODQ (10 µM) for 20 min before the application of (+)-JQ1 (10 µM). Then, contractile responses of KCl (50 mM) and PE (1 μM) were measured. (**D**,**E**): Cumulative data showing the effects of ODQ (10 μM) and L-NAME (100 μM) on the 10 μM (+)-JQ1-induced inhibition of contractility with endothelium. ** *p* < 0.01; ns: not significant; *n* = 4. (**F**): Cumulative data showing the effects of ODQ and L-NAME on the (+)-JQ1-induced inhibition of contractility in aortas without endothelium.

**Figure 3 cells-12-01461-f003:**
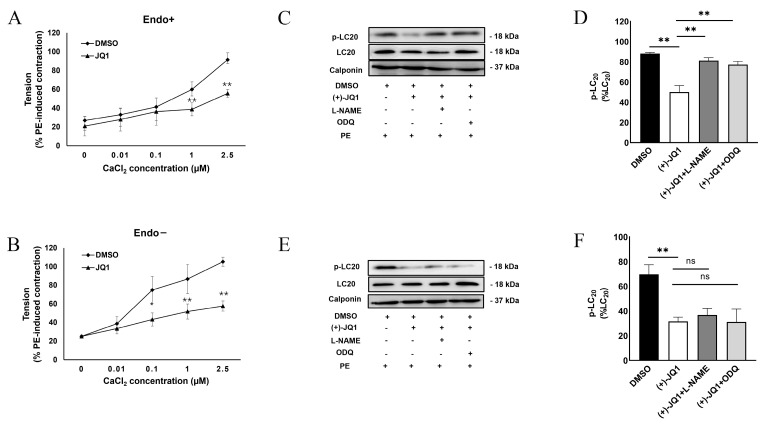
(+)-JQ1 attenuation of contractile responses is calcium-dependent, and through inhibiting the contractile machinery. (**A**,**B**): Mouse aortic tissues with (**A**) or without (**B**) intact endothelium were pretreated with DMSO or (+)-JQ1 (10 μM) and then exposed to PE (1 μM) in the indicated Ca^2+^ concentration (the *x*-axis). (**C**,**E**): Representative WB of endothelium intact (**C**, Endo+) and endothelium-denuded (**E**, Endo−). Tissues after wired myography were harvested for WB detection of phosphorylated LC20 (p-LC20) at the Ser19 site and unphosphorylated LC20 (T-LC20). (**D**,**F**): LC20 phosphorylation levels were quantified and compared to the total LC20. * *p* < 0.05; ** *p* < 0.01; ns: not significant; *n* = 3.

**Figure 4 cells-12-01461-f004:**
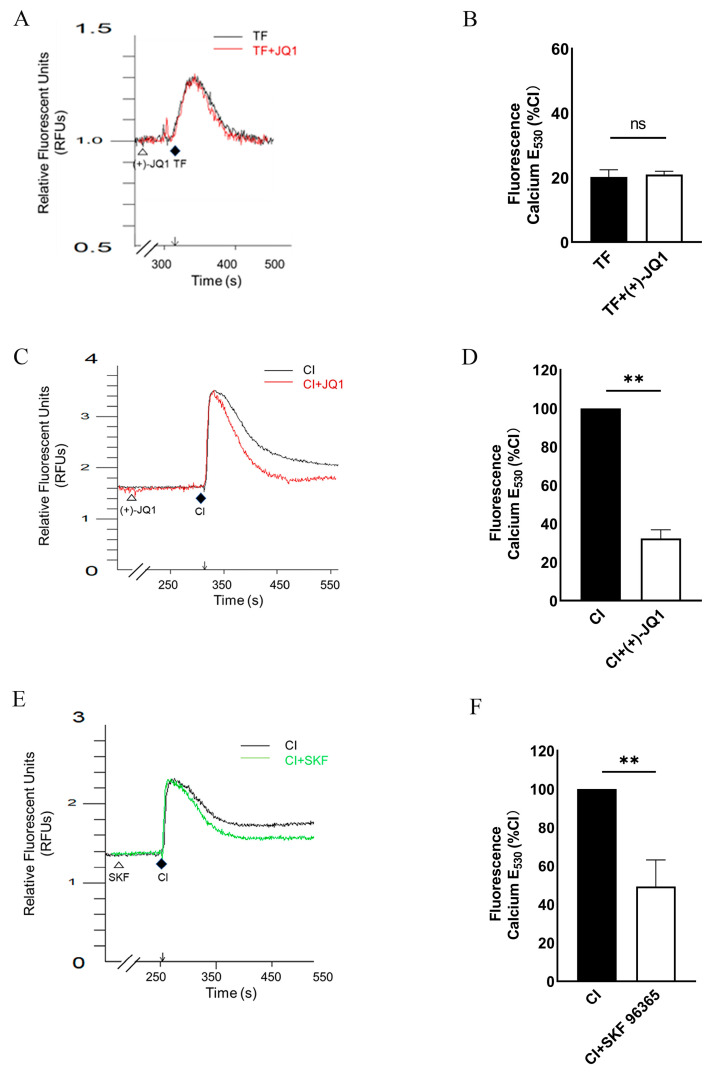
(+)-JQ1 inhibits the influx of extracellular Ca^2+^ in cultured mouse aortic SMCs. Primary cultured mouse aortic SMCs were incubated with the calcium-sensitive fluorescent dye Fura-4. (**A**): Representative traces showing that cells were pretreated with (+)-JQ1 (50 µM) or DMSO for 5 min, followed by adding 10 µM TF. (**B**): Quantification of the calcium signals as a percentage of the response induced by the calcium ionophore A23187 (CI, 2.5 µM). (**C**,**E**): 0.25 µM CI induces transient and sustained calcium increase with or without pretreatment with 50 µM (+)-JQ1 (**C**) or 20 µM SKF 96365 (**E**). (**D**,**F**): Cumulative data showing the inhibition of (+)-JQ1 (**D**) or SKF 96365 (**F**) on the influx of extracellular Ca^2+^ evoked by CI. ** *p* < 0.01, ns: not significant; *n* = 4.

**Figure 5 cells-12-01461-f005:**
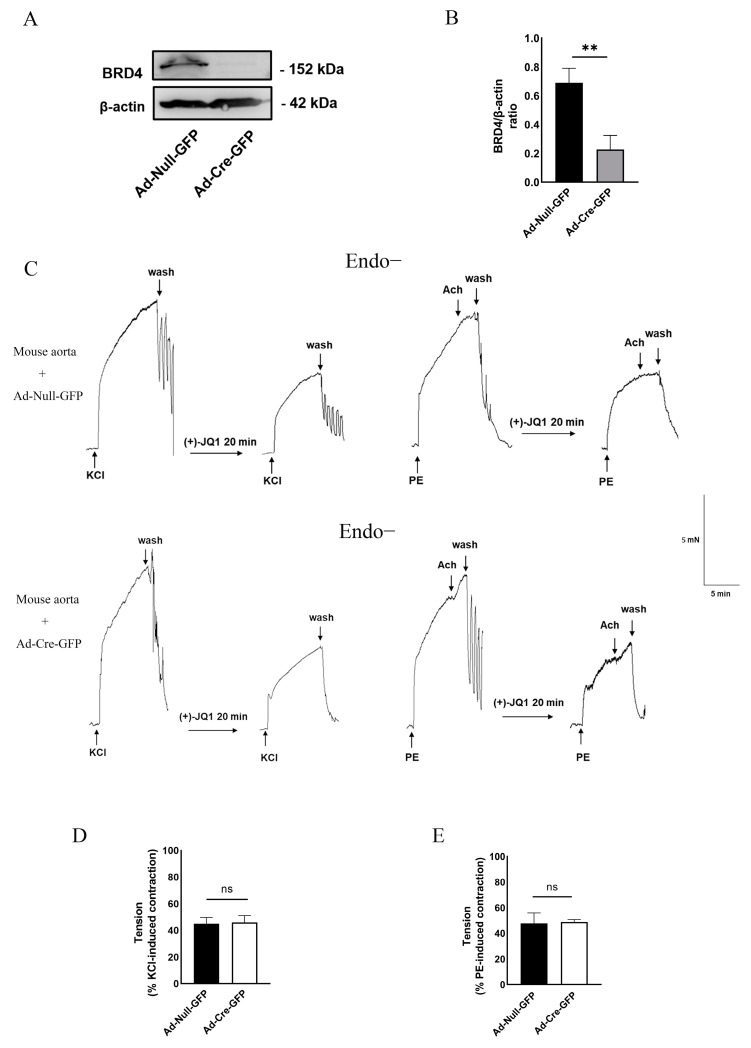
(+)-JQ1 inhibits aortic contractility of mice with SMC-specific KO of BRD4. (**A**) Tissues after the myography assay were harvested for WB detection of BRD4 and GAPDH. (**B**) BRD4 was quantified compared to β-actin. ** *p* < 0.01; *n* = 4. (**C**) *Myh11Cre/+*: *BRD4^flox/flox^* mice aortic rings were incubated with Ad-Null-GFP or Ad-Cre-GFP for 3 days, cultured tissues without functional endothelium were pretreated with (+)-JQ1 (10 µM) for 20 min, and subsequently exposed to KCl (50 mM, left panel) or PE (1 µM, right panel). (**D,E**): Cumulative data showing the inhibitory effects of (+)-JQ1 pre-contracted by KCl (50 mM) or PE (1 µM). ns: not significant; *n* = 3.

**Figure 6 cells-12-01461-f006:**
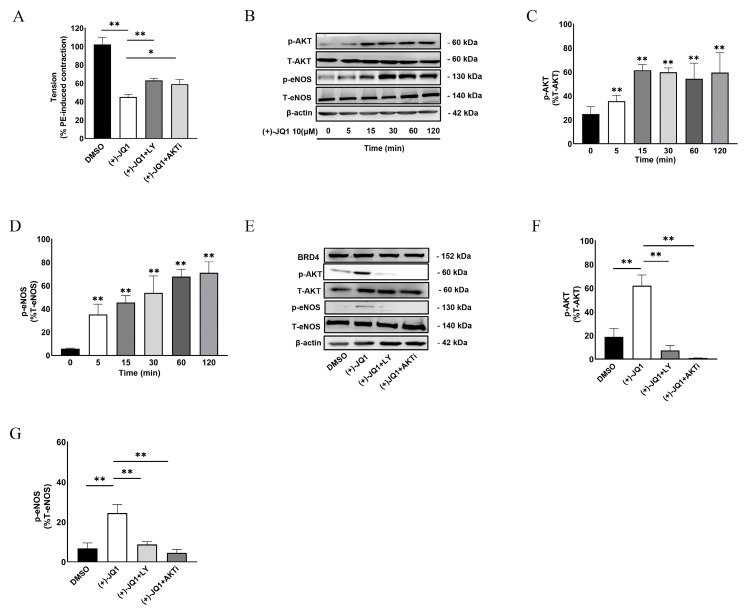
(+)-JQ1 activates eNOS through the PI3K and AKT pathway. (**A**): Mouse aortas with intact endothelium were pretreated with LY-29004 (5 µM) and AKTi IV (5 µM) for 20 min before the application of (+)-JQ1 (10 µM). Then, contractile responses of PE (1 μM) were measured. Cumulative data showing the effects of LY-29004 and AKTi IV on the (+)-JQ1-induced inhibition of contractility. * *p* < 0.05; ** *p* < 0.01; *n* = 4. (**B**): Cultured HUVECs were treated with 10 μM (+)-JQ1 for the indicated time, followed by WB detection of p-eNOS (Ser1117), T-eNOS, p-AKT (Ser473), T-AKT, and β-actin. (**C**,**D**): Summarized data showing the effect of 10 μM (+)-JQ1 on the expression of p-AKT/T-AKT ratio and the expression of p-eNOS/T-eNOS ratio in HUVECs. (**E**): Cultured HUVECs were pretreated with LY29004 (1 µM) or AKTi IV (1 µM) for 20 min followed by (+)-JQ1 (10 µM) for 30 min, then WB detection of p-eNOS (Ser1117), T-eNOS, p-AKT (Ser473), T-AKT, and β-actin. (**F**,**G**): Summarized data showing the effect of (+)-JQ1 on the expression of p-AKT/T-AKT ratio and the expression of p-eNOS/T-eNOS ratio in HUVECs. * *p* < 0.05; ** *p* < 0.01; *n* = 3.

**Figure 7 cells-12-01461-f007:**
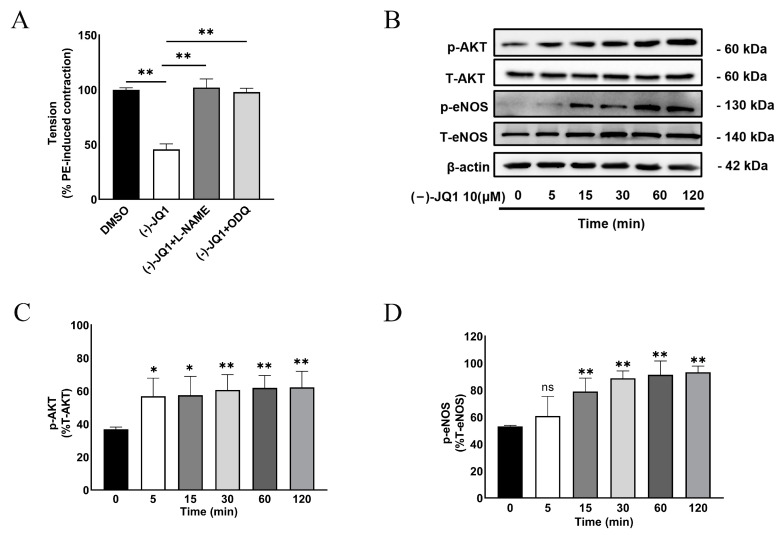
(−)-JQ1 also inhibits mouse aorta contractility and stimulates AKT-eNOS activation in endothelial cells. (**A**): Mouse aortas with intact endothelium were pretreated with L-NAME (100 µM) or ODQ (10 µM) for 20 min before the application of (−)-JQ1 (10 µM). Then, contractile responses of PE (1 μM) were measured. Cumulative data showing the effects of ODQ and L-NAME on the (−)-JQ1-induced inhibition of contractility. * *p* < 0.05; ** *p* < 0.01; *n* = 4. (**B**): Cultured HUVECs were treated with 10 μM (−)-JQ1 for the indicated time, followed by WB detection of p-eNOS (Ser1117), T-eNOS, p-AKT (Ser473), T-AKT, and β-actin. (**C**,**D**): Summarized data showing the effect of 10 μM (−)-JQ1 on the expression of p-AKT/T-AKT ratio (**C**) and the expression of p-eNOS/T-eNOS ratio (**D**) in HUVECs. * *p* < 0.05; ** *p* < 0.01; ns: not significant; *n* = 3.

**Figure 8 cells-12-01461-f008:**
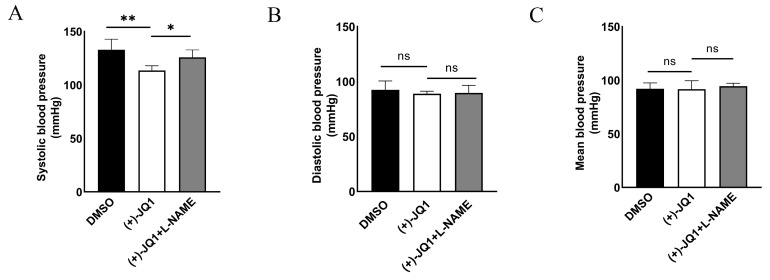
(+)-JQ1 attenuates PE-induced blood pressure elevation in mice. Mice were divided into three groups by intraperitoneal injection of DMSO, (+)-JQ1 and (+)-JQ1+L-NAME, respectively, for 14 days. Mice were injected intraperitoneally with PE (100 μg/kg). After 30 min, blood pressure was measured with the tail-cuff method, a non-invasive blood pressure system. Cumulative data showing systolic (**A**), diastolic (**B**), and mean (**C**) blood pressure. * *p* < 0.05; ** *p* < 0.01; ns: not significant; *n* = 4.

**Figure 9 cells-12-01461-f009:**
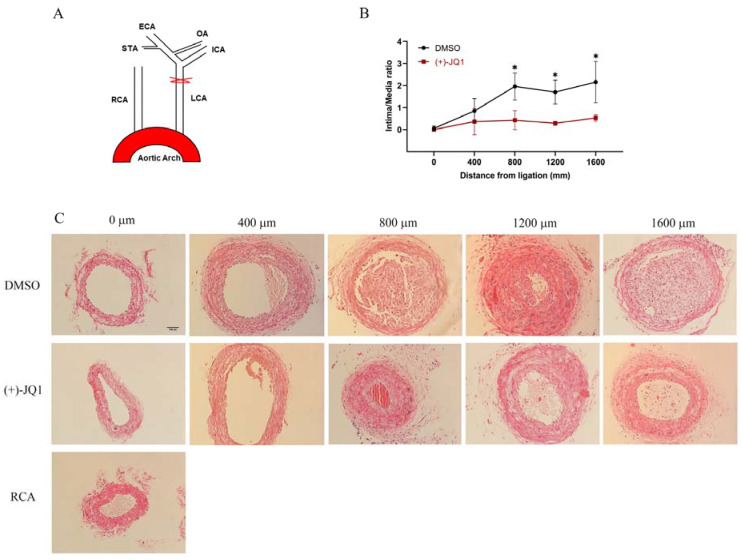
(+)-JQ1 inhibits neointima formation induced by complete carotid ligation. Male C57BL/6J mice underwent complete LCA ligation as described in the Methods section. Mice received daily treatment with 10 mg/kg (+)-JQ1 (i.p.), while the control group was treated with vehicle for 3 weeks. After 21 days, mice were sacrificed, and carotid arteries were harvested for tissue staining. (**A**): Schematic representation of complete LCA ligation, detailed in the Methods section. ECA: external carotid artery; ICA: internal carotid artery; OA: occipital artery; STA: superior thyroid artery. (**B**): Representative tissue sections (5 µm thickness) from 2 mm proximal to the ligated site were stained with HE for morphometric analyses. Neointima lesions at day 21 post-ligation with vehicle or (+)-JQ1 treatment were compared. Scale bar = 100 µm. (**C**): Cumulative data showing the ratio of intima to lumen area. * *p* < 0.05, *n* = 4.

**Figure 10 cells-12-01461-f010:**
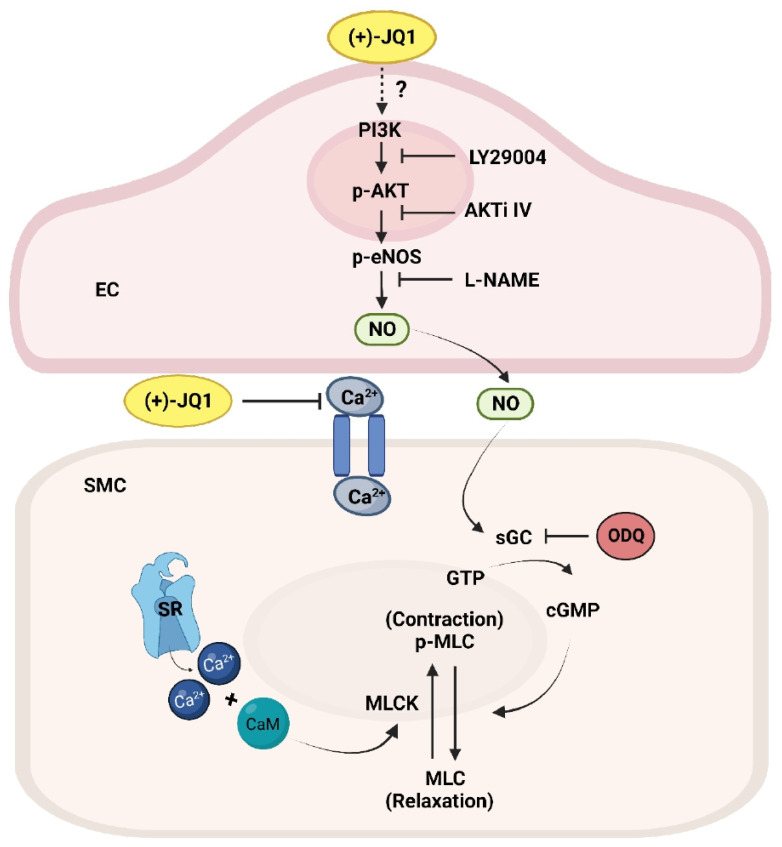
Schematic presentation for (+)-JQ1-induced inhibition of mouse aorta contractility with and without functional endothelium. (+)-JQ1 stimulates eNOS and NO production in endothelial cells through the PI3K/AKT pathway. NO activates soluble guanylate cyclase (sGC) in SMCs, resulting in an increase in cGMP and subsequent smooth muscle relaxation. In addition, the effect of (+)-JQ1 was associated with inhibition of extracellular Ca^2+^ influx, which interrupts the balance of the activities of myosin light chain kinase (MLCK) and myosin light chain phosphatase (MLCP), leading to vasodilation in the mouse aortic rings. Thus, the vasorelaxant effect of (+)-JQ1 results from both endothelium-dependent and endothelium-independent mechanisms, which is likely independent of BET protein inhibition.

## Data Availability

Data is contained within the article or Appendix A.

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
