# Peer review of "Impact of Short-Term (+)-JQ1 Exposure on Mouse Aorta: Unanticipated Inhibition of Smooth Muscle Contractility"

_cells, 2023, doi:10.3390/cells12111461_

Round 1

Reviewer 1 Report

The article is devoted to an urgent problem of modern medicine – the search for ways to inhibit the proliferation of smooth muscle cells and the formation of neointima, as well as the search for new mechanisms for regulating the contractility of smooth muscle cells. This is evidenced by a large number of publications. There are no publicly available publications that fully disclose the effect of (+)-JQ1, (a specific chemical inhibitor of the bromodomain and extraterminal family of proteins 4), as well as the mechanisms of this effect. Therefore, this publication is relevant and timely.

The authors have done a lot of work, which allowed us to establish the effect of (+)-JQ1 on the contractility of vascular smooth muscle cells. In addition, the authors have shown that (+)-JQ1 inhibits contractility of aortic smooth muscles through a dual mechanism involving activation of eNOS and inhibition of Ca2⁺ influx in smooth muscle cells.

The study was carried out at a high technical level using the latest research methods. This made it possible to use various criteria in substantiating the conclusions: functional, biochemical, morphological and analysis of the material responsible for heredity. The reliability of the results obtained is achieved by an adequately constructed design of the experiment, the use of a sufficient number of experimental animals in each group, and adequate statistical processing of the scientific data obtained.

The article is perfectly illustrated. The drawings are complete and informative. They allow unambiguously interpreting the received data.

The discussion is structured logically. The discussed provisions logically follow from the results obtained and the cited literature data.

Recommended for publishing .

Author Response

Please see our response to Reviewer 1 in our attached file.

Reviewer 2 Report

The authors from this study aimed to investigate the effects of (+)-JQ1 on smooth muscle contractility and the underlying pathways. Using several molecular and functional experimental techniques (cell culture, isolated vessels and in vivo model), the results suggest that (+)-JQ1 directly inhibits smooth muscle contractility and indirectly activates the PI3K/AKT/eNOS cascade in endothelial cells; however, these effects appear unrelated to BET inhibition. The authors proposed that (+)-JQ1 exhibits an off-target effect on vascular contractility. Overall, the manuscript is well-written, and the findings are well-supported with different experimental approaches. Several comments and questions for the manuscript is suggested below:

1) In a routine myography technique, the presence of a functional endothelium would produce robust vasorelaxation, hence generally ACH>80% is accepted to have endothelium integrity intact. In my opinion, the benchmark of ACH>50% to be accepted to have functional endothelium is considered to be rather low. How many of the vessels preparation falls in the category of <80%?

2) Given that (+)-JQ1 directly inhibits smooth muscle contractility through (PI3K/AKT/eNOS) pathway, suggesting that it could be a direct vasodilator. Did the authors perform an direct vasodilator experiments on the isolated vessel preparation? If so, is the underlying mechanisms remains to be similar?

3) Resistance arteries are important contributor to the regulation of blood pressure. Given each vascular bed is unique, it would be interesting to see if the vasodilator effects (functional antagonistic to vascular contractility) would be more/less sensitive between conduit and resistance vessels and their underlying mechanism.

4) Given the interesting off-target effect of (+)-JQ1 on vascular function, it would be interesting to test if the molecule can enhance endothelial function either alone or prevents endothelial dysfunction in various diseased conditions (diabetes, hypertension etc). This will enhance the therapeutic potential of the molecule.

Author Response

Please see our response to Reviewer 2 in our attached file.
